# Adipocyte Metabolism and Health after the Menopause: The Role of Exercise

**DOI:** 10.3390/nu15020444

**Published:** 2023-01-14

**Authors:** Megan L. Marsh, Marta Novaes Oliveira, Victoria J. Vieira-Potter

**Affiliations:** Department of Nutrition and Exercise Physiology, University of Missouri, Columbia, MO 65211, USA

**Keywords:** estrogen, visceral adipose tissue, insulin resistance, metabolism, exercise, inflammation, aging

## Abstract

Postmenopausal women represent an important target population in need of preventative cardiometabolic approaches. The loss of estrogen following the menopause eliminates protections against metabolic dysfunction, largely due to its role in the health and function of adipose tissue. In addition, some studies associate the menopause with reduced physical activity, which could potentially exacerbate the deleterious cardiometabolic risk profile accompanying the menopause. Meanwhile, exercise has adipocyte-specific effects that may alleviate the adverse impact of estrogen loss through the menopausal transition period and beyond. Exercise thus remains the best therapeutic agent available to mitigate menopause-associated metabolic dysfunction and represents a vital behavioral strategy to prevent and alleviate health decline in this population.

## 1. Introduction

The menopausal transition period is associated with increased body weight and unfavorable changes in body composition and fat deposition [1,2]. In addition to increased total body fat and adverse changes in fat mass distribution, loss of lean body mass and adipose tissue expansion and re-distribution contribute to an increased risk of metabolic disease in aging women [3]. Key metabolic changes that occur during this time include the development of central obesity and insulin resistance [4,5], which together contribute to a marked increases in the risk of metabolic syndrome (MetS), type 2 diabetes, and cardiovascular diseases among postmenopausal women [6]. Thus, the menopausal transition period is accompanied by a substantial rise in metabolic risk, which remains true even if BMI and body weight remain unchanged during this time [7]. 

While there are multiple molecular mechanisms that explain these adverse metabolic fluctuations [8,9], a combination of hormonal shifts and chronological aging are primarily what pave the way to a cluster of metabolic abnormalities associated with the menopause. Notably, physical activity levels tend to decline during and following the menopause, something that ultimately exacerbates this metabolic dysfunction. Although the extent to which physical inactivity contributes to metabolic shifts during the menopause is not fully known, it is noteworthy that the typical gain in central adiposity during the menopause is linked to an approximately 40% reduction in physical activity [10]. Comparable reductions are observed in animal models following ovariectomy (surgical removal of the ovaries to simulate human menopause), supporting evidence that ovarian hormones play a major role in these fluctuations [11]. Physical activity, however, has been shown to be highly efficacious in mitigating metabolic disturbances. Physically active women approaching the menopause tend to have a lower BMI, less fat mass, greater lean mass [12], less android adiposity, and higher bone mineral density in the femoral and spinal areas [13], and thus a lower risk of developing obesity and metabolic dysfunction during the menopause. The animal literature shows that rodents selectively bred for high fitness are protected against the metabolic dysfunction that follows ovariectomy, whereas those bred for low fitness are more susceptible [14,15]. Importantly, in both humans and rodents, increasing physical activity is an effective strategy to protect against metabolic dysfunction following ovarian hormone loss, and physical inactivity is therefore an imperative modifiable risk factor that may prevent or attenuate adverse metabolic changes during menopause [16]. Herein, we focus on the adipose-tissue-specific effects of physical activity and exercise training in this population (Figure 1).

## 2. The Menopause Increases the Risk of Metabolic Disorders

Metabolic health impacts every bodily system and is predictive of many chronic diseases [17]. Whereas optimal metabolic health equates to maintaining healthy levels of blood sugar, blood pressure, triglycerides, and a healthy waist circumference [18], MetS is a combination of risk factors, which include central obesity, insulin resistance, hypertension, and dyslipidemia [3,4]. During the menopausal transition period, changes in body composition and regional fat depots contribute to an elevation in MetS, which increases the risk of CVD and other cardiometabolic diseases [19,20]. Although younger adult women are protected against MetS, this shifts following the menopause, as we see a higher prevalence of MetS in older women compared to age-matched men [17,21,22,23]. In fact, for every 1 mmol/L increase in circulating triglycerides, there is a 76% increase in disease risk in older women, yet only a 32% increase in risk of cardiovascular disease in older men [24]. Syed et al. found that in adults aged 40–49, presumably prior to the menopausal transition period in women, men had a higher prevalence (56.7%) of MetS compared to women (42.5%) and were nearly one and a half times more likely to develop MetS than their female counterparts [25]. Mirroring this, premenopausal women tend to display a lower incidence and mortality of CVD compared with age-matched males [26], while postmenopausal women experience the opposite and are more at risk compared to older males [27]. While there is ample evidence of an increased risk of metabolic dysfunction during the menopausal transition period, it is important to note that genetic and behavioral factors also play critical roles in determining metabolic risk during and following the menopause [4,28].

### 2.1. The Menopause Increases Obesity Susceptibility

The risk of obesity increases drastically during the menopausal transition period [29,30], as evidenced by a majority of women in the United States between 40 and 59 years of age who are considered overweight or obese [31]. This is concerning because obesity is associated with a number of detrimental health outcomes, including insulin resistance [32], hyperglycemia, hypertension, hypercholesterolemia, type 2 diabetes [33], and cardiovascular disease [34]. The increased risk of obesity is linked to a decline in circulating levels of estrogen (i.e., estradiol) that occurs during the menopause. Among premenopausal women, estrogen levels associate inversely with obesity [20], which may be due to the ability of estrogen to limit the accumulation of adipose tissue [35]. In addition, estrogen loss associates with dysregulation of energy balance, although it is still unclear which side of the energy balance equation (i.e., changes in energy intake and/or expenditure) is most responsible for weight accrued during the menopause. 

While it is difficult to determine the precise cause of menopause-associated obesity in humans, rodent studies provide a highly controlled environment in which to lend insight to potential mechanisms. Estrogen-sufficient female rodents have consistently proven to be resistant to diet-induced obesity compared to males [36,37], yet this protection is lost following ovariectomy [5], suggesting that the energy balance disturbances that occur during the menopause are independent of aging. Some animal studies indicate that estrogen promotes both a decrease in food consumption and an increase in energy expenditure, thus creating a negative energy balance and decreasing the risk of weight gain and obesity [38,39,40]. Interestingly, with estrogen loss, at least in animals, reduced energy expenditure may be more important than increased intake. In some cases, there is either no change or a decrease seen in energy intake following hormone loss [41,42]. Further support for the implications of reduced energy expenditure appears in many rodent studies, which consistently show relationships between the loss of ovarian function and metabolic dysfunction (e.g., central fat accumulation and insulin resistance) that are independent of dietary changes [43,44,45,46,47]. The decline in energy expenditure during the menopause is likely due to a combination of diminished resting metabolism, loss of lean mass, and reduced physical activity levels. These changes are attributed to a combination of aging and hormone changes [48,49,50]. Highlighting the potential importance of physical activity levels per se, one group of clinical studies tracked women as they progressed from the pre- to postmenopausal period, collecting dietary records, body composition data, and energy expenditure data annually. They found evidence that the menopause-associated metabolic dysfunction observed among those women was primarily attributed to a reduction in physical activity, even after adjusting for other factors such as diet [51,52]. This work is strongly supported by the rodent literature, which consistently reveals that estrogen loss leads to reduced physical activity [53], resulting in excess fat accumulation and metabolic dysfunction. While the mechanisms for the decline in physical activity are unclear and likely multifactorial, our group is beginning to tease out the brain-specific mechanisms that may contribute to these behavioral and metabolic changes using rodent models. Our studies have implicated changes in the reward region of the brain as a potential cause of the shift in physical activity behavior [54]. 

Not only does the menopause facilitate weight gain, which directly increases obesity susceptibility, it modifies the pattern of fat distribution. Premenopausal females and age-matched males exhibit different patterns of adipose storage [55], as males tend to store adipose tissue in the visceral abdominal region while females, despite an overall higher body fat content, have a primarily gynoid distribution and subcutaneous storage of adipose tissue [56]. Ovarian estrogens promote peripheral fat storage in the gluteal and femoral subcutaneous region, whereas androgens (mainly bioavailable testosterone) stimulate visceral abdominal fat accumulation. This is irrespective of age, race, total fat mass, and other cardiovascular risk factors [28] and changes following the menopause, when adipose tissue distribution in women shifts and instead accumulates in the visceral abdominal region, coinciding with an increased risk of cardiometabolic diseases [8,29]. The observed alteration in body composition is a critical determinant of cardiometabolic disorder risk following the menopause. 

### 2.2. The Menopause Causes Dysfunctional Adipose Tissue

It is not excess fat per se that is metabolically harmful; when functioning properly, fat protects other organs, such as liver tissues, from ectopic lipid deposition. In fact, estrogen associates with increased subcutaneous adipose tissue storage, which may actually confer protection against metabolic disturbances among premenopausal women [19]. Gluteal/femoral (gynoid) obesity is associated with a reduced risk of metabolic dysfunction compared to central (android) obesity [57], a body fat distribution pattern that protects premenopausal women. However, the amount of centrally stored visceral adipose tissue in postmenopausal women can reach twice that of premenopausal women [58], with reports of central obesity three times more frequent in older women compared to younger women [59]. One study found that postmenopausal women had two-fold higher concentrations of both visceral and subcutaneous adipose tissue than premenopausal women, but only the visceral adipose tissue was linked to a more adverse risk profile [60]. That study identified blood pressure as the only variable that correlated with excess subcutaneous adipose tissue, whereas excess visceral fat was associated with a broadly negative metabolic profile. This may be explained by a higher rate of lipolysis in visceral fat due to insulin resistance. When adipose tissue becomes insulin resistant, insulin-mediated suppression of lipolysis is reduced, causing an increase in circulating free fatty acids and consequent re-distribution of these free fatty acids to the liver, ultimately promoting hepatic insulin resistance, which adversely affects whole body metabolic health [61].

In addition to increasing weight gain and shifting body fat distribution, the menopause may directly affect adipose tissue metabolism [62]. Over the past couple of decades, much has been learned about the complexity of adipocyte metabolism, with adipose tissue now considered an endocrine organ, profoundly impacted by hormones and immune cells (PMID: 25073615). Research has even progressed to elucidate how individual organelles such as the endoplasmic reticulum and mitochondria are affected, and the role they play to determine overall adipocyte health and function (PMID: 31777187). In fact, insulin sensitivity, mitochondrial function, and immune function may be considered the “triad” of adipocyte function (PMID: 25073615). There is emerging evidence that estrogen affects all three of those aspects of adipocyte function (PMID: 31868931). In the premenopausal state, estrogen tends to enhance adipose tissue insulin sensitivity, which fosters safe energy storage in subcutaneous adipose tissue and efficient insulin-mediated suppression of lipolysis. Recent work has demonstrated that estrogen loss leads to adipose tissue inflammation, which may help explain how loss of estrogen causes systemic insulin resistance. Finally, estrogen also directly affects adipocyte mitochondrial health and metabolism (PMID: 32759275), which may contribute to its roles in improving insulin sensitivity and buffering inflammation. Meanwhile, adipose tissue is the primary peripheral source of estrogen production in both males and postmenopausal females [63,64]. Estrogen influences many aspects of adiposity, including the volume and regional distribution of adipose tissue, as well as pathophysiological changes specific to adipose depot sites [65]. Generally speaking, estrogens are associated with a more favorable lipid profile compared to androgens [66]. Thus, the decline of estrogen combined with the increase in relative androgens may be mechanistically related to the adverse lipid profile seen during the menopause. Furthermore, increased androgens may diminish the positive effects of the available estrogen. Some research has proposed that the higher androgen-to-estradiol ratio after the menopause, rather than the decline in estrogen, may be linked to greater overall body fat and enhanced central adipose deposition [67]. Many reports support that the shift in this ratio is what ultimately modulates body fat quantity and distribution [9,68,69,70,71]. According to a 5-year follow-up study, a higher baseline testosterone-to-estradiol ratio and its increase over time were strongly associated with a higher risk of obesity and MetS during the menopausal transition period [72]. Another study found an independent effect of androgenic sex steroids on the adverse metabolic profile of the menopause, such that increased androgens explained the variance in visceral fat and blood lipid profile [60]. Somewhat paradoxical, higher levels of estrogen tend to associate with leanness among premenopausal women, whereas obesity increases estrogen levels among postmenopausal women, likely due to increased aromatase activity in obese adipose tissue [73]. Regardless, given that the concentrations of circulating estradiol are magnitudes lower following the menopause, the higher ratio of androgens to estrogens following the menopause is a major factor driving the adverse changes in adipose, despite elevated aromatase activity. 

In addition to the adverse effects of estrogen loss on adipocyte metabolism, obesity causes adipose tissue dysfunction, hence the combination of estrogen loss and obesity may be particularly problematic for aging women. Dysfunctional adipocyte metabolism leads to systemic metabolic dysfunction, including insulin resistance. This is because adipose tissue expansion during obesity creates a cycle of inflammation and lipotoxicity, likely due to the inability of adipose tissue to effectively store excess dietary lipids, leading to the ectopic deposition of lipids in other organs (e.g., the liver) [74,75,76]. Small adipocytes are highly insulin sensitive and protected against dysregulation of lipolysis, whereas a high concentration of enlarged adipocytes, especially in the visceral abdominal region, associates with higher fasting insulin and glucose levels [77,78]. This is partly attributed to hypertrophic adipocytes attracting macrophages, leading to an inflammatory state characterized by the secretion of numerous proinflammatory cytokines and adipokines. These inflammatory cells enter the circulation, causing insulin resistance and adversely affecting blood vessels and organs. This contributes to the greater levels of systemic inflammation observed in obesity and aging [18,79]. The failure of adipose tissue vasculature to expand sufficiently exacerbates the adverse sequelae of events that occur in obesogenic adipose tissue, which leads to local hypoxia that exacerbates the inflammatory state [80,81]. Interestingly, estrogen may be permissive to adipose tissue expandability, thus buffering obesity-associated inflammation. Mechanistically, this may be due to estrogen’s ability to stimulate angiogenesis [82] and mitochondrial activity in fat cells [83]. 

Estrogen also has anti-inflammatory effects [35] and is protective against many inflammatory disorders [84,85,86]. Mita et al. showed that obese female mice were less likely to accumulate obesity-induced proinflammatory macrophages in their adipose tissue. Male mice, on the other hand, displayed higher pro-inflammatory marker and insulin levels [35]. We and others have shown that ovariectomy in mice, independent of aging, increases the influx of inflammatory cells in adipose tissue [79]. Notably, the adipose-derived inflammation following an ovariectomy in that study was the major predictor of insulin resistance, independent of total body fat. This finding was later replicated in rat models [79,87], supporting the notion that ovarian hormone loss per se adversely affects the inflammatory state of adipose tissue. Estrogen is likely the predominant factor protecting females from inflammatory conditions, yet the mechanisms by which estrogen modulates immune function are not clear. Lumeng et al. observed a persistent accumulation of anti-inflammatory macrophages in older, obese female mice, despite a significant expansion of adipose tissue [88]. Estrogen has anti-inflammatory effects even among males, as indicated by aromatase overexpression in male adipose tissue reducing inflammation, an effect linked with improved insulin sensitivity in those mice [89]. In sum, estrogen’s protection against adipose tissue inflammation may help explain why young women are generally less susceptible than age-matched men to obesity-induced systemic inflammation. 

The protective role of estrogen extends further to insulin sensitivity [90], which may help explain why, in general, men are more insulin resistant than women. Hyperinsulinemic–euglycemic clamp studies confirm that young women have 41% greater whole body insulin sensitivity than men when matched for physical fitness [91,92]. In one study, older, obese men were found to be more insulin resistant than older, obese women, despite lower amounts of total body fat, subcutaneous fat, and greater fat free mass [93]. This also highlights that the increased risk of insulin resistance is not a result of additional fat per se, but rather excess dysfunctional fat. Despite higher adiposity relative to males, the premenopausal female body composition profile has a favorable effect on insulin sensitivity [94]. Excess visceral fat is metabolically detrimental because it tends to be dysfunctional and contributes to increased basal lipolysis, a result of insulin resistance in adipose tissue. Girousse et al. established a close association between basal lipolysis and insulin resistance, independent of BMI or age, and suggested that dysregulated lipolysis is an isolated risk factor for insulin resistance. The impact of adipose tissue function on systemic insulin sensitivity is further evident in a study of bariatric surgery patients, wherein the highest recovery of insulin sensitivity following weight loss correlated with the greatest decrease in basal lipolysis in obese individuals [95]. Park et al. demonstrated that late postmenopausal women had a reduction in insulin-stimulated glucose disposal compared to premenopausal women [96]. Whether this was due to lower levels of estrogen or excess accumulation of visceral fat was unclear; both likely contributed [97]. Research in rodents supports the implication of ovarian hormone loss by demonstrating significantly impaired glucose tolerance, higher levels of insulin, and higher resistin serum levels in ovariectomized female mice consuming a high-fat diet compared to age-matched males [5]. Nevertheless, excess visceral fat is an independent predictor of whole-body insulin resistance among males and females [28] and associates with both skeletal muscle and adipose tissue insulin resistance. Despite its strong relationship with insulin resistance in both sexes, estrogen may safeguard visceral fat in metabolic health. Research indicates that visceral fat from ovary-intact females is more insulin sensitive than that from obesity-matched males, as determined by increased protein levels of phosphorylated Akt and ERK in response to insulin stimulation [98]. Further support for a protective effect of estrogen on insulin sensitivity is an observed estrogen-mediated increase in insulin sensitivity in males, whereas a complete lack of estrogen synthesis or activity in men is associated with insulin resistance [99,100]. 

### 2.3. Role of Adipose Tissue Estrogen Receptors in the Link between the Menopause and Metabolic Risk

The mechanisms by which estrogen is metabolically protective likely involve estrogen receptor (ER) signaling [101,102]. Thus, the mechanistic investigation of how estrogen affects adipocyte metabolism begins with an in-depth understanding of its receptor-mediated actions. Both major subtypes of the estrogen receptor, alpha (ERα) and beta (Erβ), are heavily expressed in adipocytes and mediate estrogen’s effects [102]. These receptors have both similar and distinct actions upon estrogen binding; thus, selective ER modulators offer a way to target the tissue-specific effects of estradiol (E2). Acting via its nuclear transcription factor receptors, estrogen plays an important role in altering genes that regulate adipogenesis, lipogenesis, and lipolysis. Estrogen’s ability to stimulate adipocyte lipolysis, which allows for greater fat oxidation, may also be attributed to its receptor-mediated actions [103]. 

When local estrogen production is inhibited via knocking out aromatase, animals become obese and insulin resistant. A similar phenotype is caused by selective knockout of ERα [47,104], suggesting that ERα may metabolically mediate the protective effects of estrogen. In fact, several preclinical studies in rodents have demonstrated that estrogens activate protective effects against insulin resistance through activation of the ERα pathway in insulin-sensitive tissues [104,105]. Unfortunately, the adverse risks associated with hormone replacement therapy (HRT) are almost completely driven by the actions of E2 via ERα [106,107]. Thus, recent work implicating the metabolic benefits of the ERβ-mediated effects of E2 [108,109] is very exciting. Emerging data indicate that ERβ ligands have adipocyte-specific metabolic benefits, including obesity reduction. 

## 3. Exercise Improves Metabolic Health Following the Menopause

The benefits of exercise are broad and affect virtually all body systems. Across populations, higher levels of physical activity are associated with greater attenuation of body fat gain and preservation of lean mass across the lifespan [110]. Indeed, lifestyle activities that include regular physical activity can effectively improve metabolic health and prevent MetS [12,111,112]. Evidence shows that sedentary, overweight, and obese postmenopausal women are very responsive to exercise training, so physical activity is a highly beneficial behavioral practice for improving overall health in this population. Exercise is particularly effective at reducing visceral fat and thus critical in mitigating the accumulation of visceral fat during the menopause. In addition to facilitating fat loss, exercise is also likely to improve fat cell metabolism and may be effective in “replacing” the beneficial effects of estrogen [113]. In a study examining groups of postmenopausal women with similar weight loss, only the group that exercised experienced a favorable shift in body composition with a reduction in the android to gynoid fat mass ratio [114]. Research has also shown a dose–response effect of exercise on fat loss and metabolic parameters [115,116]. Earnest et al. examined how different “doses” of exercise affected metabolism and found, remarkably, that significant improvement seen in waist circumference, fasting glucose, and systolic blood pressure were dose-dependent, once again highlighting the therapeutic effect of exercise, especially among aging women [117]. Gonzalo-Encabo et al. conducted an analysis of pooled data from two randomized controlled trials and assessed the dose–response effects of aerobic exercise on adiposity markers in a total of 720 postmenopausal women. Both trials spanned one year and randomized participants to low (150 min/week), mid (225 min/week), or high volume (300 min/week) exercise. All of the exercise groups improved their fitness levels, as indicated by improved VO_2_ max scores, and higher exercise volumes resulted in statistically significant reductions in BMI, weight, fat mass, and fat percentage. Interestingly, weight loss among the exercise groups was not significantly different, although there was a trend for a dose–response effect. The effects of exercise dose on body fat percentage were greater among women randomized to the low-volume group who already had a low BMI. However, among women with a high BMI, there was a trend for a greater reduction in total fat mass with increasing exercise volume, and the weight loss experienced among those women was due to a loss of fat while lean mass was maintained. Most importantly, the authors observed a greater reduction in intra-abdominal and subcutaneous abdominal fat in conjunction with a greater exercise volume, the first study demonstrating this effect in postmenopausal women [115]. Another study showed a dose response related to exercise intensity, such that the ability of exercise to reduce visceral adiposity appeared to be driven by exercise intensity. That is, vigorous intensity exercise induces greater reductions in waist circumference compared with moderate intensity [118], thus supporting the effectiveness of exercise to selectively target visceral fat in a dose-dependent manner.

In studies using rodents, rats bred to have higher life-long levels of physical activity, and consequently a higher fitness capacity (measured via running capacity), are protected from the metabolic dysfunction that accompanies ovarian hormone loss via ovariectomy [119]. Conversely, rats bred to have low levels of physical activity show an exacerbated response to adverse metabolic changes following ovariectomy [11]. Nevertheless, in both rodent groups, exercise training effectively mitigated insulin resistance and other signs of metabolic dysfunction [96]. The powerful influence of exercise on whole body insulin sensitivity in an intensity-dependent fashion is demonstrated as glucose uptake by working muscles rises 7 to 20 times over its basal level during physical activity [120]. Since metabolic health, and adipose tissue metabolic health in particular, deteriorates after the menopause [121,122,123], postmenopausal women benefit tremendously from regular exercise. Furthermore, the benefits of exercise extend beyond metabolic improvements. One study investigated the impact of aerobic exercise coupled with dietary restriction on hormonal, metabolic, and psychological variables in postmenopausal women. The combination of exercise and diet produced significantly more beneficial effects than diet alone on a variety of outcomes. Improvements were seen in serum levels of sex hormones, insulin resistance, and depression scores compared to the non-exercise control group [124].

### 3.1. Effects of the Menopause on Responsiveness to Exercise and the Role of ERs

While the numerous benefits of increased physical activity are clear across populations, the mechanisms through which the menopause affects responsiveness to exercise must be considered. Ample evidence indicates that exercise improves fat loss among postmenopausal women when compared to diet-only strategies, yet some studies have suggested that, compared to premenopausal women, postmenopausal women may be partially resistant to the adipose tissue-targeted effects of exercise. Other research has revealed that postmenopausal women may have impaired exercise-induced fat oxidation, making exercise-mediated weight loss more difficult. Abildgaard et al. examined the effect of fat oxidation in a cross-sectional study on pre-, peri-, and postmenopausal status and its influence on the oxidative capacity of skeletal muscle via skeletal muscle biopsies taken before and immediately after an exercise bout, in addition to measurements of indirect calorimetry during exercise. Postmenopausal women had significantly lower whole body fat oxidation and energy expenditure during exercise than the premenopausal women. Furthermore, postmenopausal women demonstrated a blunted exercise-induced increase in phosphorylation of AMPK, which coincides with the suppression of fat oxidation during exercise. The authors ultimately concluded that the reduction in whole body fat oxidation after the menopause is likely attributed to a reduction in lean body mass [3]. Further studies in postmenopausal women have shown that lean women have greater fat oxidation during exercise than obese women. Upon measuring the lipid accumulation index and maximum fat oxidation during submaximal exercise, women with a high lipid accumulation index had 33% lower whole body fat oxidation and 19% lower energy expenditure during exercise than those with a lower lipid accumulation index [125].

Both sex and age have been shown to affect adipose tissue ERs and adipocyte metabolism. Porter et al. compared omental (visceral abdominal) and abdominal subcutaneous adipose tissue genes and proteins in a total of 28 premenopausal women, 16 postmenopausal women, and 27 age-matched men undergoing bariatric surgery (i.e., all obese). With the exception of fasting non-esterified fatty acids, which were higher in women, no other differences were found in other indicators of systemic glucose and lipid metabolism. Upon examining the gene and protein levels of the ERs, omental adipose tissue ERβ levels were higher in older women than in younger women and older men, and aromatase expression was higher in older men than in older women. Overall, among the patients studied, the subcutaneous adipose tissue immunometabolic profile was heavily influenced by both age and menopause status, more so than omental adipose tissue [126]. Subsequently, Ahmeda et al. conducted a similar study and also found that subcutaneous adipose tissue in postmenopausal women had lower expression of ERβ than premenopausal women. The data showed that ERβ expression levels in subcutaneous fat correlated inversely with central obesity and positively with adipocyte glucose uptake. Furthermore, primary adipocytes extracted from subjects in that study were treated to knockdown ESR2 (the gene encoding ERβ) and it confirmed a reduction in glucose metabolism in the absence of ERβ [127]. 

It is unclear how changes in the ERα and ERβ density of adipose tissue affect glucose in lipid metabolism or how such changes may affect exercise-mediated fat oxidation. However, in the study by Porter et al. described previously, the protein content of both ERα and ERβ was highly correlated with the mitochondrial protein uncoupling protein 1 (UCP-1) across sexes and ages [126]. Those same authors, using preclinical models, have demonstrated that both exercise and a selective beta 3 adrenergic receptor ligand, CL316,243, increased the density of ERβ in both subcutaneous and visceral fat, suggesting that stimulation of adipocyte metabolism may cause increased ERβ expression on adipocytes [128,129]. Meanwhile, a series of preclinical studies in rodents have shown that selective activation of ERβ has metabolism-boosting effects [108,109,130]. Thus, capitalizing on adipose tissue ERβ expression following the menopause may improve metabolism by mitigating both weight gain and insulin resistance. In fact, research has shown in genetically mutated mice, that in the absence of ERβ, the adverse metabolic effects of estrogen loss are worse [131]. Specifically, when ERβ was absent, ovariectomy led to more severe obesity, the accumulation of visceral fat, and metabolic dysfunction. Bridging this back to whether changes in ER expression may affect exercise responsiveness after the menopause, a preclinical rodent study revealed that ERβ-mutated mice were resistant to the adipose-tissue-specific metabolic adaptations with exercise that were observed in wild-type mice [132]. Intriguingly, both exercise [133] and the adipose-tissue-specific “exercise mimetic” CL316,243 [53,129,134] increase ERβ density selectively in adipose tissue. Capitalizing on this enhanced ERβ expression, particularly following hormone loss, combining selective ERβ activation with exercise (or drugs that affect adipose tissue metabolism) may be an effective strategy to maximize maintenance of metabolic health following the menopause. 

### 3.2. Exercise-Modality-Specific Effects

#### 3.2.1. Strength Training

Due to the strong influence of aging and the menopause on body composition, including the loss of lean mass, exercise designed to improve and maintain muscle mass is critical. In this regard, strength training has been the focus of some studies examining how exercise may mitigate metabolic dysfunction following the menopause. A study investigating the association of resistance training exercise frequency and volume with changes in body composition among postmenopausal women found that resistance training was successful in preventing weight gain and deleterious changes in body composition. The authors determined that training frequency was significantly and inversely associated with changes in body weight, adiposity, and trunk fat (a marker of visceral fat). This was true even after adjusting for age, years on hormone therapy, changes in lean soft tissue, baseline body composition, and baseline habitual exercise [135].

Since loss of estrogen is at least partially responsible for the accompanying metabolic dysfunction observed during the menopause, it is possible that combining hormone replacement therapy with resistance training may be even more beneficial than exercise alone. Exercise and estrogen therapy may synergize to improve metabolism in this population. One study determined the effects of resistance training with or without transdermal estrogen therapy (ET) on adipose tissue mass and the metabolic risk profile in early postmenopausal women. Here, untrained postmenopausal women were allocated to supervised resistance training with a placebo or transdermal ET for 12 weeks. Body composition, metabolic-health-related blood markers, body fat percentage, adipocyte cell size, and lipogenic markers in subcutaneous adipose tissue from the abdominal and femoral regions were assessed. While resistance training improved those outcomes, the ET group unexpectedly experienced a lesser reduction in total and visceral adiposity compared to the placebo group. However, the ET group improved their metabolic blood profile as indicated by reduced low-density lipoprotein, glucose, and hemoglobin A1c [136].

#### 3.2.2. High-Intensity Interval Training

The metabolic benefits of high-intensity interval training (HIIT) are increasingly well established. A recent meta-analysis concluded that HIIT is a time-efficient, feasible, and effective intervention to modify body composition by reducing abdominal and visceral fat mass [137]. Accordingly, HIIT may also be an effective approach to treat and prevent metabolic dysfunction following the menopause [138]. Dupuit et al. compared moderate-intensity continuous training or HIIT training with or without resistance training to evaluate improvements in the body composition of postmenopausal women. The results indicated that body weight and total fat mass decreased in all of the groups over time, but HIIT training was significantly more effective at reducing abdominal/visceral fat mass. Additionally, while the combination of HIIT and resistance training did not potentiate this effect, it led to an increased percentage of muscle mass [139]. Lean mass increases the basal metabolic rate and total energy expenditure, so the combination of HIIT and resistance training may be an equally beneficial exercise modality for postmenopausal women. In another study investigating the most effective approaches for weight loss among postmenopausal women, Grossman et al. compared HIIT to endurance exercise for 16 weeks. Remarkably, participants randomized to the HIIT group lost twice as much weight as those in the endurance training group (8.7% vs. 4.3% of initial body weight lost, respectively), and only the individuals in the HIIT group experienced significant improvements in body composition (e.g., lost fat and maintained lean mass) [140]. 

Some studies have suggested that the effects of HIIT may be diminished following the menopause. In fact, the Dupuit study described above determined that HIIT was significantly more impactful in reducing weight through total and abdominal adiposity prior to the menopause [138]. Nonetheless, HIIT is still an appropriate strategy to limit CVD risk in postmenopausal overweight/obese women. In a recent randomized controlled trial examining the effect of HIIT training on endothelial function and hemodynamics in postmenopausal women, He et al. compared HIIT to moderate-intensity continuous training for 8 weeks. While both interventions improved endothelial function, HIIT was particularly effective at enhancing flow-mediated dilation as well as vasoconstrictive and vasodilatory responses. These results allowed the authors to classify HIIT as a feasible training protocol for improving endothelial function among postmenopausal women [141]. Another study investigated the effects of HIIT on the inflammatory and adipokine profile in postmenopausal women with MetS before and after a 12-week training period. The authors reported improvements in circulating inflammatory markers and positive changes in circulating adipokines. Overall, the data displayed an anti-inflammatory effect of HIIT, along with fitness and body composition improvements. Whether the anti-inflammatory effect was a direct consequence of training or the result of other physiological improvements was undetermined [142].

#### 3.2.3. Alternative Forms of Exercise

In a randomized controlled trial assessing the effects of a 16-week functional fitness protocol utilizing elastic resistance bands in conjunction with balance and agility exercises, significant improvements in all body composition variables related to fat were observed among the postmenopausal women studied. Significant improvements in total cholesterol and HDL were also observed [143]. Thus, functional training utilizing elastic bands and unstable bases appears to significantly improve body composition, functional fitness, and lipid profiles in postmenopausal women. 

One study looked at tai chi as a means to improve the adverse metabolic symptoms of the menopause. The authors investigated the effects of a 12-week training regimen and saw improved body composition, muscle strength, functional capacities, blood pressure, and general health perception in postmenopausal women [144]. Furthermore, after only 12 weeks of tai chi, these women experienced a significant decrease in their waist circumference and significantly lower systolic and diastolic blood pressures. Thus, tai chi and potentially other forms of meditative exercise may be a beneficial in promoting metabolic health in postmenopausal women.

Similar studies have looked at the effects of yoga on menopausal symptoms and sleep quality. Some studies observed significant decreases in menopausal symptoms in both peri- and postmenopausal women following 20 weeks of participation in yoga classes, with greater effects in postmenopausal women. Additionally, sleep quality improved among all of the participants, indicating that yoga can improve health and wellbeing during the menopausal transition period through multiple mechanisms [145]. A different study compared the effect of yoga and regular exercise on psychophysiological improvements in postmenopausal women. Researchers found that the yoga group scored better compared to the exercise and control groups in questionnaires that evaluated climacteric syndrome, stress, quality of life, depression, and anxiety. They concluded that yoga has positive effects in the psychophysiological changes experienced by postmenopausal women [146]. 

## 4. Nutritional Approaches to Improve Metabolic Health Following the Menopause

While there is ample evidence in both human and animal models that weight gain occurs following the menopause and the loss of ovarian hormone production, whether this metabolic disturbance is due to increased energy intake through dietary changes or reduced energy expenditure is still unclear. The MONET study, which followed a group of women through the menopausal transition period while annually tracking dietary patterns and energy expenditure, found that observed weight gain was attributed to reduced physical activity and total energy expenditure and not increased energy intake [147]. Studies in rodents confirm these findings, consistently demonstrating that ovariectomized rodents gain weight due to a reduction in energy expenditure, specifically through a decrease in spontaneous cage activity and voluntary wheel running [15]. Regardless, reduced energy expenditure, with or without increased dietary intake, contributes to weight gain and metabolic dysfunction during the menopause. 

Whether or not dietary patterns change during the menopause may be a difficult question to address, but new research may shed some answers. Decades ago, during the Women’s Health Initiative (WHI), a dietary intervention study was implemented to determine the effectiveness of diet in mitigating postmenopausal health impairments. Postmenopausal women were instructed to adopt a low-fat, high-fiber diet for a 9-month period and through a moderate reduction in energy intake, they weighed ~2.2 kg less on average than those in the control group by the study’s end. Notably, in addition to consuming less fat, the intervention group consumed more fiber, vitamins and minerals, dairy products, and fish [148,149]. Along with the many healthful dietary modifications observed, postmenopausal women saw a reduction in ER+ breast cancers that approached statistical significance. The findings also revealed the minor protective effects on total deaths and diabetes that were strongest in women who were obese at the beginning of the trial [150,151]. Later, from 2004 to 2006, owing to the limitations of self-reported dietary analyses, the WHI expanded its trials to include biomarkers, specifically precise assessments of energy expenditure along with urinary nitrogen to validate protein consumption. Then, between 2010 and 2014, a feeding study was conducted to identify the important biomarkers of dietary patterns and how such patterns influence health and disease status [152]. These more sophisticated analytic studies demonstrated a clear correlation between energy surplus (mediated through excess energy intake or reduced energy expenditure) and increased adiposity, metabolic dysfunction, and, ultimately, increased disease risk. As noted in a recent review on the history of nutrition-focused research conducted under the WHI 18, the future nutritional epidemiology research agenda needs to include both self-reported and biomarker measures of nutritional intake and disease status as well as metabolomic profiles [153]. Such studies may lend critical insights into the best strategies for dietary modification to improve health among our growing population of postmenopausal women. 

Research has also shown the benefits of exercise in modifying gut microbial populations [154]. A recent study involving a large cohort of postmenopausal women revealed important interactions between hormonal, physiological, and dietary factors that may synergize as determinants of metabolic risk [155]. Interestingly, data has shown that the gut may play an important mediating role in predicting risk among postmenopausal women. As a part of the ZOE PREDICT project, this was a large cohort study that compared pre- and postmenopausal women with age-matched men. Controlling for factors such as age and BMI, the results identified the independent effects of the menopause on aspects of metabolic health, supporting prior studies, such as those described herein. Novel findings have highlighted the adverse effects of the menopause on postprandial metabolism and glycemic control, as measured using continuous glucose monitors. In addition, the menopause undesirably affected sleep quality and added sugar intake (12% worse sleep and 12% more added sugar). Regarding HRT, favorable associations were found for visceral adiposity as well as fasting and postprandial metabolic indicators. Statistical analyses have revealed new and potentially significant associations between diet, gut microbial changes, and health indicators [155]. Importantly, this study highlights the profound impact of diet and lifestyle protections for aging women and supports a closer look at HRT, especially for women with high susceptibility for metabolic disease who do not have contraindicated health risks, such as a history of reproductive cancers. We can ultimately conclude that various combinations of diet, hormone therapy, and exercise may be extremely beneficial in optimizing whole body health throughout the menopausal transition period and beyond. 

## 5. Conclusions

Women spend up to three decades in the postmenopausal phase of life, which is often accompanied by a wide variety of disruptive manifestations, including vasomotor symptoms, sleep disturbances, cognitive decline, mood disorders, and metabolic dysfunction [156]. While aging coincides with weight gain, shifts in body composition, and metabolic dysfunction in both sexes, the abrupt alterations in hormone levels associated with the menopause coincide with a stark increase in metabolic risk [156], leading to an alarming rise in cardiovascular disease and diabetes prevalence among postmenopausal women. Research has consistently demonstrated that adipose tissue becomes metabolically dysfunctional following the menopause and is causally related to other concomitant conditions and diseases, such as breast cancer [157] and non-alcoholic fatty liver disease [158]. Irrespective of whether this increased cardiometabolic risk is more heavily driven by behavioral changes or hormonal shifts, human and rodent data alike provide clear indications that both exercise training and adopting or maintaining healthy dietary patterns following the menopause are essential in mitigating visceral fat accumulation and preserving metabolic health. Thus, preventing weight gain and adipocyte/metabolic dysfunction during the menopause is crucial, and exercise is currently the best tool available. 

## Figures and Tables

**Figure 1 nutrients-15-00444-f001:**
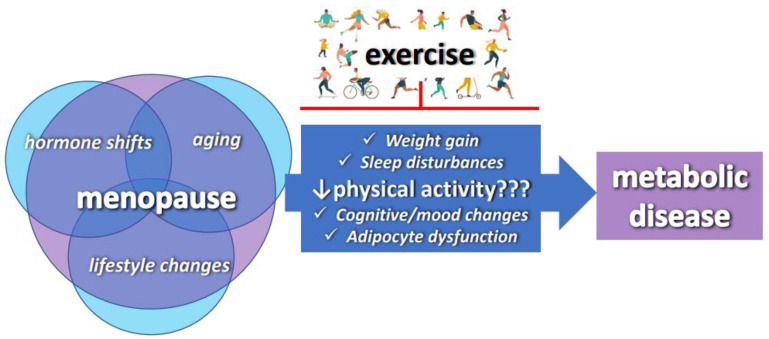
Schematic representation of the complex etiology of menopause-associated metabolic dysfunction and the hypothetical role of exercise in mitigating menopause-associated metabolic changes.

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
