# Peer review of "Adipocyte Metabolism and Health after the Menopause: The Role of Exercise"

_nutrients, 2023, doi:10.3390/nu15020444_

Round 1

Reviewer 1 Report

The manuscript presents a comprehensive review of the metabolic effects of menopause, based on updated literature. In fact, the entirety of manuscript content is probably known by specialized physicians and reseachers. Since readers of Nutrients have different profiles, the manuscript would gain in clarity and readbility if authors include some schematic ilustrations, preferably relating how the different physiologic phenomena associate to menopause are afffected by the interventions mentioned, such as diet, exercise, etc.

Author Response

Dear review 1,

Thank you for taking the time to read our manuscript and for your thoughtful and helpful insight. I completely agree with you regarding the state of knowledge and diverse readership. I have constructed a schematic diagram illustrating the main thesis of the manuscript to help readers understand the broad overview. Here I attach that new figure.

Reviewer 2 Report

This is a very well-written, comprehensive review of the state of the science related to exercise as a therapeutic intervention for the adverse metabolic effects of menopause. The investigators are experienced researchers in the field of exercise physiology. 

As a reproductive endocrine researcher, I only have one minor criticism: both the abstract and the text claim that estrogen loss is associated with reduced physical activity, implying a hormonal effect. The evidence for this is mostly in rodents and the two reports they cite in humans are from the MONET study which doesnt specifically demonstrate corroborative evidence. It could be argued that indirect socio-behavioral factors in middle age, rather than estrogen loss, might be to blame. Given that rodents dont experience menopause per se, and the evidence in humans is far from conclusive, I suggest a softening of the language in both the abstract and the paragraph in the manuscript starting on line 105. examples below/

ABSTRACT: In addition, estrogen loss  MAY BE associated with reduced physical activity,

The decline in energy expenditure during menopause is likely due to a com-bination of diminished resting metabolism, loss of lean mass, and PERHAPS reduced physical activity levels. These changes are attributed to a combination of aging and hormone changes 104 (68, 75, 137).

Having said that, this idea is indeed interesting and certainly worthy of further investigation, as the investigators have acknowledged. As part of the abstract, this could even be re-framed and highlighted as a hypothesis. 

Author Response

Dear Reviewer 2,

Thank you so much for your positive comments and helpful feedback. I appreciate very much your taking the time to review our work and am happy that you think it will be a positive contribution to the field. Thank you for your comments regarding physical inactivity as part of there etiology of menopause-related metabolic dysfunction. Indeed, this is a hypothetical argument and we were too strong it our statements regarding this. We have constructed a figure to better portray the hypothetical model (attached here) and will revise the text within the document as well. 
